# Non-Spatial Data towards Spatially Located News about COVID-19: A Semi-Automated Aggregator of Pandemic Data from (Social) Media within the Olomouc Region, Czechia

**Jakub Konicek** , **Rostislav Netek \*** , **Tomas Burian** , **Tereza Novakova and Jakub Kaplan**

Department of Geoinformatics, Palacký University in Olomouc, 17. listopadu 50,
77146 Olomouc, Czech Republic; jakub.konicek@upol.cz (J.K.); tomas.burian@upol.cz (T.B.);
tereza.novakova02@upol.cz (T.N.); jakub.kaplan01@upol.cz (J.K.)
**\*** Correspondence: rostislav.netek@upol.cz

**Abstract:** The article describes the process of aggregation of media-based data about the coronavirus pandemic in the Olomouc region, the Czech Republic. Originally non-spatially located news from different sources and various platforms (government, social media, news portals) were automatically aggregated into a centralized database. The application "COVID-map" is an interactive web map solution which visualizes records from the database in a spatial way. The COVID-map has been developed within the Ad hoc online hackathon as an academic project at the Department of Geoinformatics, Palacký University Olomouc, Czech Republic. Alongside spatially localized data, the map application collects statistical data from official sources e.g., from the governmental crisis management office. The impact of the application was immediate. Within a few days after the launch, tens of thousands users per day visited the COVID-map. It has been published by regional and national media. The COVID-map solution could be considered as a suitable implementation of the correctly used cartographical method for the example of the coronavirus pandemic.

**Keywords:** COVID-19; location based data; social media; aggregator; map application

## 1. Introduction

The world was paralyzed by the coronavirus pandemic in the first half of 2020. Under these circumstances, various media and information to inform the public about the current local and global situations spontaneously emerged through different channels. However, this rapid development often resulted in disinformation. Society was overloaded with duplicate information from dozens of sources with differing quality and this may have adversely affected public opinion. Sufficient information was available for the Olomouc Region of Czechia, but in the first stages of the pandemic, it was fragmented, inconsistent and produced by many different sources (social networks, news portals, official crisis staff sources, personal profiles of the governor and mayor, etc.).

The motivation behind the COVID-map application was to aggregate information from various sources into a single location and to verify these reports. Crisis management during the first stage of the pandemic did not have any defined unified communications strategy. Much information was published through unofficial channels, such as personal social networks. The objective with the application was to concentrate all necessary information about the region into one place and indicate information on a map. Social responsibility and dissatisfaction with incorrect applications of basic cartographic methods in various alternative solutions also motivated the present authors to create the COVID-map.

The article presents and verifies the process of aggregating non-spatial information into a map during the crisis. The project's backbone was defined during a 24-h hackathon at the Palacký University Olomouc's Department of Geoinformatics. Several updated versions were deployed that followed a workflow of agile development. This allows us to respond more effectively to both users' feedback and the pandemic situation. The aim was to produce an application which could effectively visualize originally non-spatial and thematically focused reports on the coronavirus published by various sources. The article describes the principles of the aggregator for data which were originally non-spatial in character and verifies the applicability of visualizing current reports on a map.

## 2. Motivation

### 2.1. General Motivation

The COVID-19 disease, which Europe was able to witness through events in China at the beginning of the New Year, spread dramatically around the world within a few months. The infection has gradually grown from a local epidemic into a global pandemic. The disease, which is caused by a new type of beta-coronavirus SARS-Cov-2, has not yet been fully described by science. Unfortunately, no vaccine is currently available. But how is the coronavirus pandemic different from others, such as Ebola, SARS or MERS? For a long time, these viral pandemics have created adverse effects throughout the world. The current situation was highlighted by strong commercialization through some sections of the media, which other reports kept to a minimum. Governments imposed very strict restrictions, such as quarantines, prohibitions of movement, the closure of schools and restaurants, etc. The economy and associated international trade also slowed down. The world began to stop. This health crisis will most certainly cause a financial crisis, resulting in global economic recession.

### 2.2. Technological Motivation

Quarantine has forced many people in the world to remain at home. With the spread of the pandemic, the need for modern technology and communications equipment has become more pressing than ever. Social distancing has created a rapid increase in communications and work via the Internet. School education, business meetings, theater performances and many other events are now taking place in the form of online meetings. As a consequence, people are looking more for information about the pandemic and home restrictions. For this reason, tens to hundreds of websites have been created for the purpose of providing information about the current numbers of infected, recovered and deaths. The very first successful site was a dashboard with a map showing the number of cases of infection. The map was created by the Johns Hopkins University in the USA (described in Section 3.2). Other similar web pages and map applications have subsequently appeared. This is a prime example of the practical use of Geographical Information System (GIS), whose advantages are fast data processing, modern visualization methods and a relatively wide range of presentation options for the general public [1]. While GIS is a complex software for spatial data input, analysis and visualization [2], map is an output generated by GIS. This mainly concerns spatially localized information; however, the spatial component contained in most media reports does not achieve its full potential described only in text. Its potential is much greater, and the information value is higher when information is visualized graphically and indicated on a map.

### 2.3. Social Responsibility

The rapid spread of COVID-19 has taken the world by surprise. The Czech Republic established strict restrictions and declared a state of emergency when the virus appeared in the country. Universities, shops, restaurants and other establishments were closed at a moment's notice. The government prohibited the free movement of people throughout the Czech Republic, and mandatory quarantine commenced. At this point, many people understood the consequences of the situation. Society began to ask questions about this new type of coronavirus. What were its symptoms? Was a medicine or

vaccine available? What was the number of infected cases in the Czech Republic? Had anyone nearby been infected yet?

Unfortunately, fake news spread across the Internet and media. The main idea behind the project was to provide the public with information about current events in the Olomouc region related to COVID-19. Awareness in the population of relevant news has been a key aspect in calming the tense situation and encouraging compliance with the announced restrictions, which has ultimately contributed to limiting the spread of the virus. The aim was to aggregate all important reports into a single site where users could easily find all the information they needed. Adding a spatial component promoted knowledge of the geographical aspect of news and helped to create a complete picture of the issues surrounding the pandemic. By aggregating sources, users could find information in one place and did not need to search through dozens of other news portals. Verification of the sources ensured the veracity of the news.

## 3. Research of the Current Conditions

### 3.1. Impact of Social Media on Crisis Management

Social networks have become a part of our everyday lives. Because many contributors are willing to share their own more or less personal data, we can observe a huge accumulation of differing information. Two of the best known and most used social platforms worldwide are Facebook and Twitter. Contributors to these social networks use hashtags based on keywords in their posts. Hashtags enable posts or statuses to be grouped according to topic. It is not unusual to be able to extract information only a few hours, sometimes even just minutes, after any event and use the data. The data consist of brief information or eyewitness accounts of certain incidents [3].

Besides social networks, several desktop and mobile applications are designed for crisis management. One of the best known is Ushahidi, which means 'testimony' in Swahili, originally made to help monitor political upheavals in Kenya. Each user with an internet connection can supply information about current situations and assist rescue services. Ushahidi was also used during the earthquakes in Haiti (2010) and to help victims during the terrorist attacks in Christchurch [4,5]. Its use stands against the background of VGI (Volunteered Geographic Information), which is a type of crowdsourcing. In the field of geoinformatics, VGI has helped to create a participatory GIS, which refers to obtaining, managing and handling spatial information and provides information about living environments to disadvantaged groups and communities [6].

### 3.2. About COVID-19 Map Applications

Since the COVID-19 pandemic began, many map applications have been designed to monitor the disease. Solutions have been created through open-source communities, commercial enterprises, universities and health ministries.

One of the most used and shared coronavirus mapping applications is TrackCorona, made by Stanford University in the USA. In addition to visualizations, the Stanford application offers information about the disease and prevention tips. A useful feature is its geolocating tool, which allows people to discover how many positive cases are in the user's area of interest.

Johns Hopkins University produced a monitoring application using the Esri platform [7]. The application includes a global overview along with numbers of infected, deaths and recovered patients. The US data also include the numbers of tests conducted and laboratories involved. The Swiss Ministry of Health also used the Esri platform to create a Swiss coronavirus application. Both the Swiss and US applications were made in the Dashboard App by Esri [8]. In this context, Esri shared their methodologies and tips for unconventional visualizations on their ArcGIS blog. A full walkthrough is available and takes users through the selection of map coordinate systems and projections to the creation of relevant maps using coxcombs and how to interpret them [9].

German newspaper Berliner Morgenpost created a web map application which mainly focused on Germany, although broader European and global data may also be found. The map is only available in German and is based on open source technology Leaflet, OpenStreetMap and MapTiler [10]. It also includes an animated timeline of the development of the pandemic. Coronavirus data is also available for the Shires of the United Kingdom, showing a number of graphs, maps and tables [11]. However, the choropleth map in the background of the maps has been incorrectly applied, since it only displays absolute numbers of patients.

The Canadian government created a simple web page summarizing important information about the coronavirus situation in Canada. The application publishes tips and advice for Canadian citizens according to groups at risk. Canadian data can be downloaded as a CSV file [12]. A visually very interesting map application was created by developer Andrew Jackson in New Zealand, using New Zealand Ministry of Health data. The application combines MapBox and OpenStreetMap data [13].

### 3.3. Data Agregators

Data aggregation is a data mining technique which summarizes data (sometimes called 'big data') into statistics for larger spatial units. The amount of data creates collections which can later be used in batches. The main aim is to use aggregators for direct data analysis [14].

Data aggregators generally take the form of intelligent software which reveal relationships between data attributes. Attributes are first defined by users according to preferences which can be used individually or be combined. This also offers a means of filtering data, since only selected data is used. Data aggregation is important across many fields, including geoinformatics, finance, tourism and industry.

Transnationally, data aggregators are used quite often. A European Commission through Eurostat built the GISCO (Geographic Information System of the Commission) system. It integrates raw data in the database, adding the geospatial character determining a location and providing spatial-oriented visualization [15]. It can be considered as an explicit example of GIS usage for transformation for a variety of different kind of data. Three simple steps—localize, analyze/verify, visualize—describe this research as well. A Statistical Commission of the United Nations introduced The Global Statistical Geospatial Framework. One of the main goals of the framework is to ensure the consistent aggregation of geospatial and statistical data [16]. Various tools and software can be used to aggregate data, but aggregation may also be executed manually. Automated aggregation is performed with middleware (third party software, plugins or tools), for example Improvado, Periscopa, Data or Birst. The COVID-map application described in the present paper uses an automated non-spatial data aggregation method based on a plugin. A technical description of the solution is detailed in Section 4.1.

## 4. Materials and Methods

### 4.1. Input Data Sources

The application fully automates the process of collecting information found in different formats and platforms from numerous data sources. It combines both spatially localized and non-spatial information originating on different platforms (social networks, news web portals, official crisis management sources) in various data forms (statistical data, text data, multimedia) and file formats. Input data sources are therefore a basic prerequisite in publishing and verifying the data. Because it is a web-based solution, input data sources are available exclusively through the Internet protocol. They contain the following types of data:

- Non-spatial data—news reports from web portals (7 sources—automatically)
- Non-spatial data—posts from social networks (14 sources—automatically)
- Statistical data—for the whole region and cities (1 source/6 data—automatically)
- Spatial data—map layers (3 sources/5 layers—manually).

The most important input layer is original information in the form of reports or posts imported from several types and sources. These reports are not spatially localized upon entry and contain primarily news from established news servers: one local and two regional newspapers, publishing services, regional television, regional radio, national television and university news. All eight sources provide a standardized RSS format for news subscription. They are imported into the application via an RSS feed. The second, very valuable type of information is obtained from social networks. Social media such as Facebook and Twitter generally provide breaking news quicker than any other channel. The private profiles of the regional governor and mayors have been especially valuable and immediate sources of information. The social networks category includes 14 different profiles, such as the official Facebook channels of the Fire and Rescue Services (the basic crisis staff units) and public transport office, the official Facebook profiles of cities across the region, Facebook news channels and the private profiles of the governor and mayor (Facebook and Twitter). Importing is fully automated via the Facebook API. The most common type of information requested by the public is the current statistical data, which typically concerns the number of people who have tested positive for the coronavirus or number of people recovered. Information for the entire region and each city is available separately. The Crisis Staff of the Olomouc Region provides statistical data on behalf of the open data strategy. Automated and digitally processable data are available through a specially designed webpage: https://www.krajpomaha.cz/. Open data are published in the standardized and easily-processable JSON file format. Importing is again fully automated. The application implemented six statistical data metrics: number of positively tests, number of recovered persons, total number tested, ratio of positive/tested, ratio of positive/number of people for the entire region and number of positively tested for individual cities. The final type of information is spatial data, which is available directly in the map as layers for regional borders, the boundaries of municipalities closed for quarantine, police and military stations established during quarantine to hermetically close an area (source: Fire rescue services), closed border crossings (source: www.mapy.cz) and sampling points (source: https://www.krajpomaha.cz/).

Importing all the above-mentioned types of data is fully automated; only spatial data requires manual implementation in the map application. However, due to the static nature of the data, it is a disposable operation. All map layers use the GeoJSON format. Currently, the platform combines four types of data from different sources imported through different formats. It is designed as an open source solution ready for further extension.

*4.2. Technology*

From a technological point of view, the web application was developed fully according to modern standards and adapted and optimized for display on mobile devices (responsive design). The solution is available online at the URL https://gis.upol.cz/covid/.

Hardware

The shared server at the Palacký University data center in Olomouc contains the peak of hardware architecture. The server parameters enable development of map applications or web presentations according to modern standards: Linux operating systems, Ubuntu distribution, php 7.0.33, Apache 2.4.18 web server and SSL support for HTTPS. The complete server specification is available at http://gis.upol.cz/info.php. The parameters of the server allow a capacity of up to 50,000 impressions/15,000 unique users per day. Its limit is around 240 simultaneous unique visitors every second during peak hours (see Section 4.3). All descriptive information, attributes and metadata are stored in MySQL database 5.7.29.

Selecting suitable software was crucial to the development process (see Section 4.3). The objective was to launch the application as quickly as possible and then improve it iteratively, involving students in the development process. The original requirement was to create a universal administrative platform for content and user management that could communicate effectively with the map library. Its core

does not meet proprietary solutions strictly tied to a single technology. Based on the input analysis, several requirements essential in the choice of technical solution were identified: interoperability, an open source principle enabling fast connection between various libraries, future scalability using agile development and the choice of an established CMS (Content Management System) without needing to develop a custom backend solution.

The application was developed using a combination of the WordPress + Leaflet solution. The WordPress 5.3.3 platform provides the backend and the non-map section of the frontend, i.e., the homepage and news list, in a non-map form. The fundamental technical component of the entire system is the principle of data aggregation from various sources (Figure 1). For this step, the "WordPress Automatic Plugin" was selected. The plugin allows various data types (RSS, HTTP, social networking APIs Facebook, Twitter, Instagram, YouTube, etc.) to be imported from any number of sources. The map interface is operated by the Leaflet 1.4.1 library, and the map layers are in the GeoJSON format only. This combination was selected for the following reasons:

- Leaflet is user-friendly, easy to develop and possesses a well-documented library which fully meets the needs of a mapping solution.
- Routine knowledge of the Leaflet library and GeoJSON format is held by students [17], which enabled immediate deployment.
- In general, the choice of CMS minimized the time and effort for backend development and native functionality (user access and rights, post sharing, import/export, etc.),
- WordPress is a ready-to-use platform for verifying reports, with the option of broad customization.
- WordPress as a data aggregator—a plugin available for WordPress which enables fully automated import from various sources and completely eliminates the need for manually processing data collection.

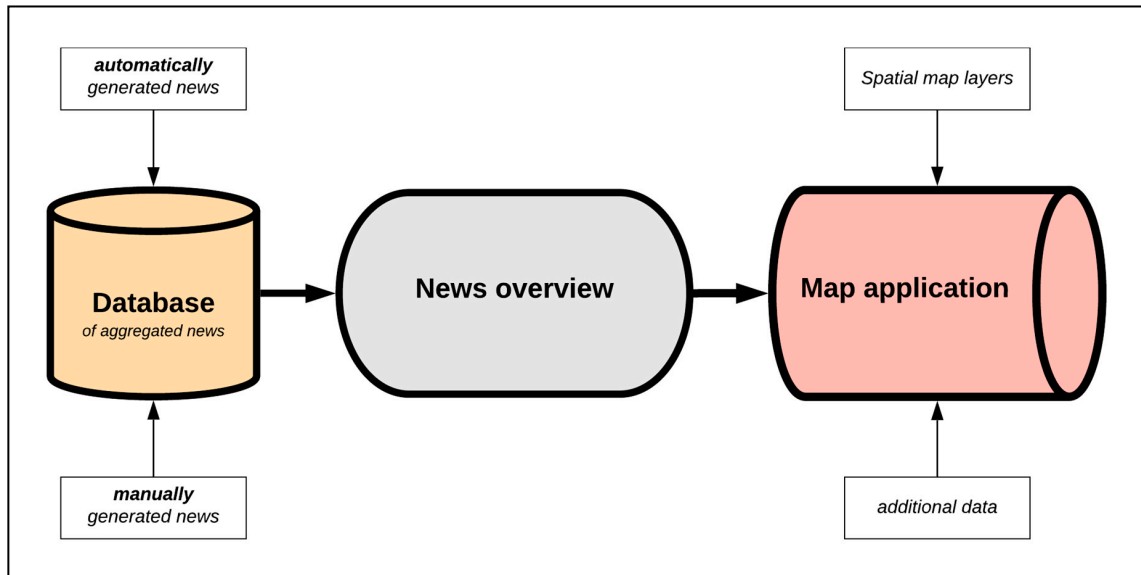

**Figure 1.** Technological schema of the COVID-map project.

*4.3. Project Workflow*

Creating the map application was a gradual process and influenced by continual changes to improve the content and form of the final public product. To minimize the time for the entire initiative, most of the work for improvements was done live. The individual processes and work of the implementation team are described in the following chapters.

### 4.3.1. Administration

Most of the administration team consisted of students from the Department of Geoinformatics at Palacký University Olomouc. Nine students, a PhD student and an assistant professor were involved. Each member was assigned a specific role (Public relations, front-end development, design, verify) according to their knowledge and preferences. These roles were given based on Design Thinking process models for the most effective process [18]. An unambiguous schedule for the verification process was set up for efficient deployment and elimination of duplicate personnel and empty time windows. Communication among members was centrally maintained, and everyone was able to intervene in the creative process with comments and observations that assisted in detecting errors and potential shortcomings. During the work, the cartographic themes of Prof. Vít Voženílek (Vice-President of the ICA) and Dr. Alena Vondráková (senior cartographer at the Department of Geoinformatics) were addressed. Their recommendations helped optimize the cartographic representations on the map and resulted in cartographic rules being strictly followed and the creation of correct, easy-to-read and non-misleading visualizations. Online platform Slack was used as central platform for organization and communication; Google Drive was used as the documents storage. Online calls via Skype and Microsoft Teams were held to eliminate any physical meetings. Thematically, this reduced any potential spread of the coronavirus (not only) in the Olomouc region by the team.

### 4.3.2. Agile Development

The initiative to create a map application arrived when the coronavirus crisis appeared in the Czech Republic and multiple government restrictions had been applied. The situation changed very rapidly, and it was important to respond promptly not only during the creative process but also during development of the product itself.

The virtual hackathon provided an opportunity for creative thinking and to involve students with different levels of knowledge. With considerable time pressure due to a very quickly and dynamically changing situation, agile creation and management methods were applied in full respect to effective design thinking process. Agile software development is an interactive project management methodology based on the continuous delivery of a product or service and active cooperation with users who provide continuous feedback for small improvements [18,19]. This solution was an ideal choice for the COVID-map creative process. According to the methodology, the best usable artefacts (tools) and appropriately compiled events (time management) were defined for each team member assigned a role (Section 4.3.1).

Agile development roles are divided into Product Owner, Scrum Master and Team [20]. The Product Owner's role represents the customer. This member monitors the development of work and ensures that the final project concepts are followed. The creator of the initiative acts as the Scrum Master. This member sets research questions, and demonstrates the concepts of the final product. The team fulfils tasks according to a set of clear instructions and are consulted at regular intervals via video conferencing. Project creation using agile development is characterized by three basic steps [21]: planning, cooperation and delivery.

The plan, is a complete list of tasks to be done at the end of the project. Each task receives a priority which designates when the task must be fulfilled (higher priority requires quicker completion of the task). Requirements are set in advance. When they are fulfilled, the task is designated as complete, i.e., the definition of done (DOD) [20]. Inspired by this knowledge, a shared document was created to establish the main points of the project: Thematic areas of interest; Project concept design; Internal architecture of the application; Map content and processing; and Sources of information.

Several tasks to fulfil the final application were defined for thematic areas of interest (i.e., what might interest the user). The tasks included stimuli for visualizing information about the numbers of infected and recovered persons, spatial distribution of the infected, closed areas, sampling points for testing for infection and aggregation of information sources from local authorities.

Inspiration from the already existing COVID-19 map applications assisted in designing the layout in the Adobe XD environment (Figure 2). A functional prototype could then be created. The default materials for creation are described in Section 3 in a detailed overview.

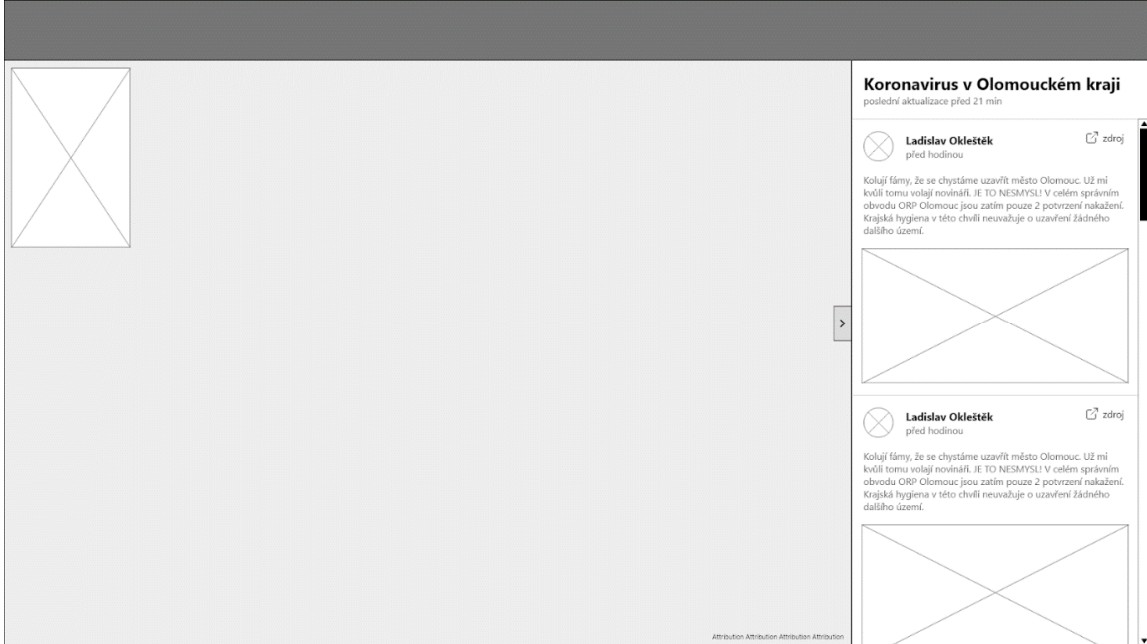

**Figure 2.** Schematic layout of the COVID-map.

The design of the internal architecture, and therefore also the entire web environment, was guided as much as possible by the principles of minimalism and simplicity. Each user can access the information intuitively. Divisions were restricted to the introductory page, which served as a signpost to the next two pages: a map application and a page which dynamically generated news as text from information sources. Each of these sections was supplemented with notes and inspirational screenshots from selected existing solutions.

The information contained in the map needed to be visualized correctly, easily and effectively. The plan therefore included cartographic visualization methods to show the number of infected, closed areas, and information sources in the region. The forms for graphically processing base maps, the color design and the sources of available layers subject to free use were also determined. The final point concerned the policy of freely available data, which is a current topic in the Czech Republic on account of poor availability and quality. The whole cartographic domain of map creation is described at Section 5.2.

Obtaining a wide range of information sources was not a difficult task given the huge media coverage of COVID-19 issues. All information sources were properly structured according to location and character (news and information from state authorities) and subsequently grouped according to whether they could be processed automatically or manually (see Section 4.1). A total of 55 information channels were collected and actively used. According to Panek et al. [22] collaboration is an iterative process (a 'sprint') involving a repeated, limited unit of time. The team discusses the project's workflow and any potential changes, and shortcomings can be subsequently incorporated and improved. A sprint review of the project was conducted in several identification stages, which led to progressive improvements. In the first few stages, the implementation team acted out the role of a customer. At the end of the hackathon, application users provided feedback and tested full operations. Debugging was done during the full start-up.

The first test version was produced in 24 h using simple HTML, CSS and JavaScript code. The map contained a layer of administrative units and test data. The layout and basic functionality of the tools in the designed environment were tested. The first form of the map application was divided into two basic sections, a sidebar and map, and compiled according to the proposed concept in the planning stage. After the first iteration, the public version was launched on the university server at the URL https://gis.upol.cz/covid/. Accordingly, the process of generating the initial channels in the side panel commenced. In this iteration, the map was populated with layers of municipalities in the Olomouc region which had been ordered into quarantine and closed, point closures for roads and border crossings, and ORP administrative units in the GeoJSON format. Figure 3 depicts the current map visualization, which was supplemented by simple switching between the available layers.

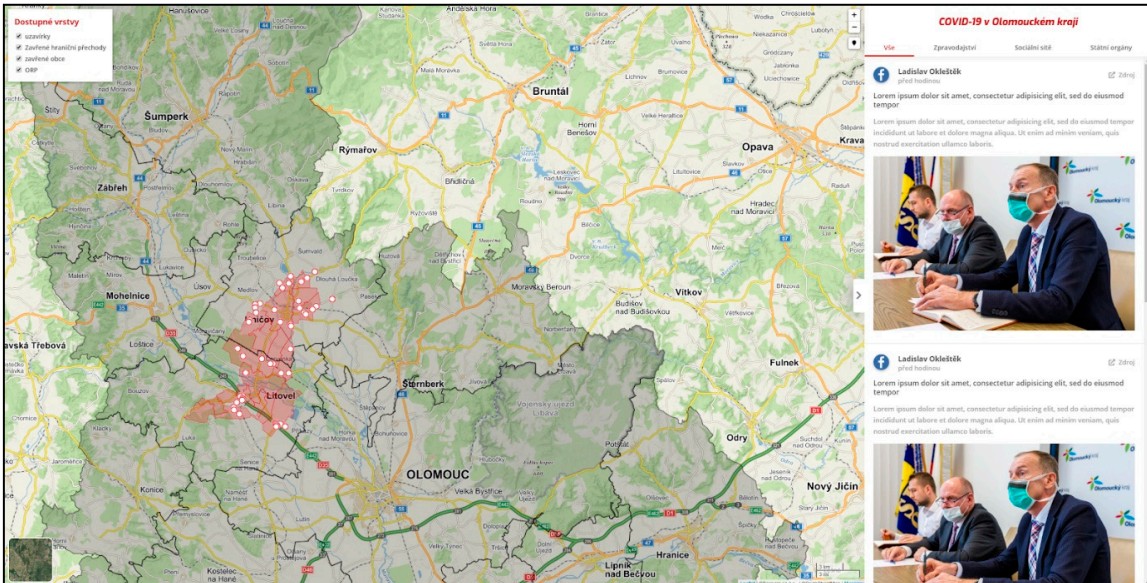

**Figure 3.** The first public version.

Error debugging targeted the entire agile application development process and was executed in small steps. Changes to individual elements were always executed en masse. The primary stage focused mainly on eliminating problems in functionality and ensuring user-friendliness. The next steps were eliminating cartographic deficiencies and increasing the application's level of graphics.

The main challenges included optimization and therefore visualization of the news in the map window so that information would be clearly assigned to the region and clearly distinguishable. The volume of published information increased continuously, and it was not possible to display information in groups of thematically selected point characters. It was necessary to aggregate points into clusters which automatically allocated information into a single, thematically visualized point symbol which related spatially to a municipality with extended powers [23]. The numeric indicator described the number of reports for a given area. Upon user interaction, the cluster divided into smaller sub-units representing specific reports. Figure 4 depicts the progress from the initially developed version to the current version. The legibility of the map itself and clarity of the visualized data clearly represents the feedback obtained from active users.

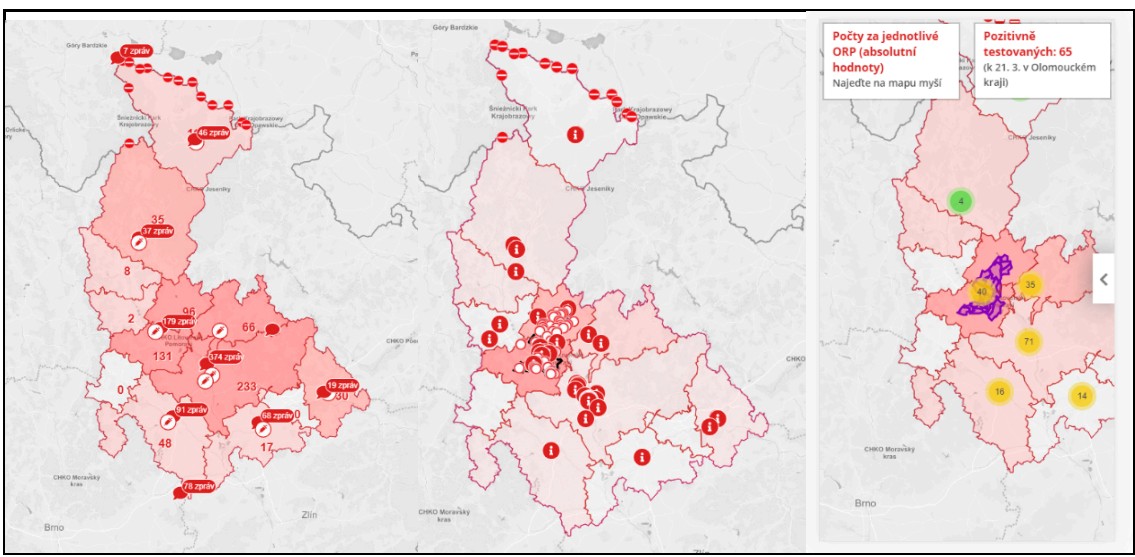

**Figure 4.** Visual evolution of the clustering method.

According to web statistics, more than half of users accessed the application from a mobile device. Therefore, after launching and resolving the major functional deficiencies in the pilot version, the responsive version for mobile phones was optimized over several iterations. However, the main components and concept behind the project were successfully optimized for mobile devices. Figure 5 indicates the gradual development of the interface for mobile devices from a de facto empty map field into a fully functional version.

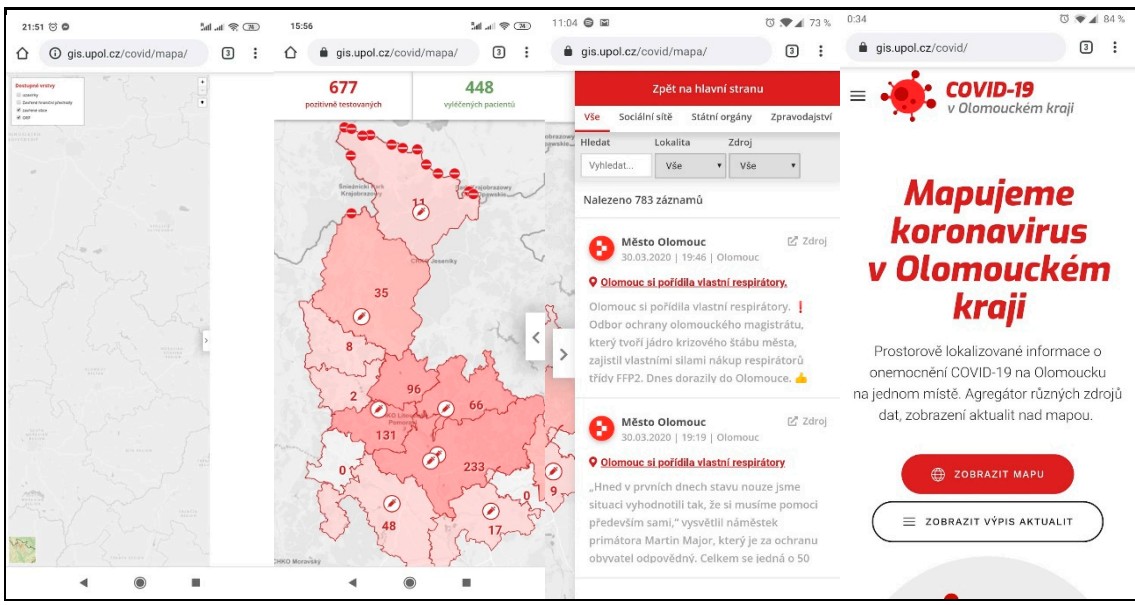

**Figure 5.** The mobile version of the COVID-map.

### 4.3.3. Data Processing and Spatial Verification

The overall process of publishing data is semi-automated. The objective in this defined process is to verify and publish information as quickly as possible and follow a near real-time paradigm. As mentioned in Section 4.1, two types of non-spatial data are automatically entered into the database system via RSS and the Facebook API.

Based on the selected technical solution and the specific workflow, the first step in the aggregation process is importing data from defined sources (see Section 4.2). Importing is fully automated according

to an hourly schedule. Greater efficiency in data verification is achieved by distributing all the sources between several verifiers, i.e., certain sources are allocated to the responsibility of specific individuals. After successfully importing the data, these individuals are notified by email that the source has been allocated to them. At this point, posts are fully imported into the form of a native WordPress post without any spatial characteristics to await verification.

In the next stage, each post is manually verified, and its spatial location determined so that it can be correctly assigned into the map application. The verification process excludes posts which do not thematically or locally fall within the project's aims. This stage also eliminates any typographical or other textual errors which are undesirable for presentation (e.g., incorrect and missing Czech diacritics). Fully automated publishing was tested in the first stage of the project, but unfortunately demonstrated an unsatisfactorily high error rate (approximately 30%) in filtering data. The validation process affects map interpretation by users directly. In fact, verification is the filter of visualized and non-visualized news within the map. Therefore, a selection of students for verification was crucial. A university level of education provides a minimum level of critical thinking and responsibility. Only students with a local background were selected. Collective training took place with the aim to set the same verification rate and rules at the very beginning of the pilot study. The choice of imported sources was made collectively under the supervision of senior lecturers. Only official sources were included. For example, social network profiles of the governor and mayor are private but marked as official. In fact, social media treated them as hot news as, while other classic media used them as a primary source as well. These scheme of validation supports information duplicity, when one topic is shared by several media sources. That was specific case for the most important news (hermetically close area, first death etc.) or nationally valid news. Clear duplicity was eliminated by verifiers. Statistical information was imported automatically from the only official source provided by Crisis Staff, no mismatching was possible.

The second stage of the process is geolocation, i.e., the spatial delimitation of each contribution. Based on the information from the source, title, keywords or content of the report, each verifier unambiguously assigns a spatial location to the post with point coordinates. This step is supported by entering the address into a form with an autocomplete feature (geocoding process) or by directly inputting coordinates (by placing icons on the map). The coordinates are stored in the database in WGS84. It is a standardized format of two parameters "lat,lng" (latitude, longitude) for each post. The geocoding process is executed with the Nominatim library, a native geocoding tool available under the OpenStreetMap project. The address entered in the form is sent through the API as a query containing three basic parameters: searched address name ("Olomouc"), output format ("JSON"), definition of the geometry type ("point"), for example: https://nominatim.openstreetmap.org/search? q=olomouc&format=json&point=1.

This specific example outputs code available after opening the provided link. The API returns the result in the JSON format with the parameters "lat" and "lon", which are then stored in the database. On the client side of the application, both spatial and non-spatial data are then generated from the database into a map interface using a PHP script [24].

Other up-to-date statistical data are taken directly from the crisis staff website https://www.krajpomaha.cz/. As an official source of regional crisis management staff, the site provides all data in the JSON format. Sources are selected according to the greatest relevance and accuracy in terms of spatial and thematic focus, while its data are updated at regular intervals, guaranteeing their timeliness. A time unit indicator for statistical data validity has also been explicitly placed in the map application to prevent misinformation. The update interval is set for every hour, followed by data storage into a database. Other map layers are static and do not require updating.

## 5. Results

The web environment of the COVID-19 application can be divided into three basic units, consisting of an introductory page, which provides a signpost between the main results of the project, the map

application and a generator to spatially describe aggregated information sources. These individual components are assembled graphically and simply in a friendly interface to ensure easy and intuitive access for a wide range of users with different technical skills. The following chapters describe the front-end solution of the final project. The description focuses on the concept behind the application and cartographic processing.

*5.1. Front End (UI/UX)*

The primary interface is the home page at the URL https://gis.upol.cz/covid/. It has an informative and orientative function and enables easy access to the main map application and aggregated news list. The website's simple architecture intentionally dispenses with unnecessary add-ons. The chief informative value is given to the other two outputs.

The introductory page's interface is built according to an intuitively designed template which respects the requirements of a responsive website. The top panel contains a thematically created logo and links to the sub-pages of the aggregator's architecture. The page's most prominent feature is the main field, which provides information about the focus and core of the project with a brief educational description. Below the textual information are two prominent and graphically different icons referring to the map and list of news. The graphically different footer contains a more detailed description of COVID-map objectives in the Olomouc Region and explains how to implement it. Important links and useful contacts are highlighted. Essential information is also displayed, including telephone numbers for the Ministry of Health and Regional Hygiene Station, the website of the State Institute of Public Health and thematically focused information from Palacký University Olomouc.

The map application respects current trends in creating web cartographic products. The layout (Figure 6) is divided into three fully responsive, mandatory components containing a permanent information panel at the left, a news panel at the right and the interactive map itself.

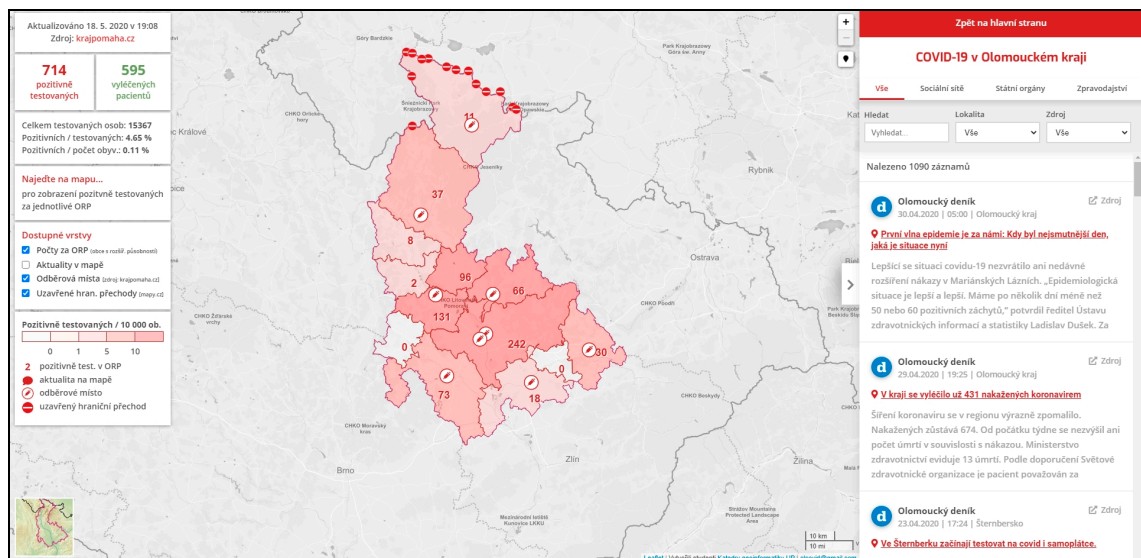

**Figure 6.** Layout of the map application.

The permanent information panel has several informative sub-functions providing numerically and thematically colored data on the current number of people who have tested positive for the coronavirus and statistics on the number of recovered patients. Data are supplemented with statistics of the total number of people tested, the ratio of positive cases to the number of tested people and the number of inhabitants in the Olomouc Region. Presented in this manner, the statistics represent an unbiased view of the current state of the disease across the region. User-friendly and accurate information transfers are indicated in a window, which displays the precise values for a specific area and visualizes the statistics according to where the cursor is moved over the map.

Another function in the left-hand panel switches the map layers between relative and absolute numbers of infections for a specific area. Point layers indicate sampling locations, closed border crossings and news in the map representing the number of aggregated reports for a given area. The lower section contains a legend which clearly explains all the elements located on the map. The main feature of the entire interface is an interactive map with a fixed bounding box positioned on the Olomouc region. Because the visualization continuously focuses on the area of interest, users are not able to search for other locations. For the underlying map, "mapbox.light" by Mapbox is used by default. The toggle in the lower-left corner allows switching to a general geographical background map produced by Mapy.cz. The currently active layers are visualized in shades of red over the base map. The selection of appropriate base map was fundamental. The base map directly affects data interpretation by users–inappropriate designed background (e.g., without labels or an indistinct road network) makes users' orientation difficult. It could also lead to incorrect spatial localization of thematic information. The map field is complemented by zoom tools, localization using the user's IP address, map scales and descriptions containing references to data sources along with links to contact information for authors.

The right-hand panel is a smaller version of the sub-page showing the specific aggregated outputs of all information sources. A link at the top also allows users to return to the main page. The panel is enhanced with two methods of individual filtering reports. The first allows sources to be displayed according to the origin of their publication, i.e., from social networks, state authority websites or news channels. By default, it is set to display all reports without any preferences. Advanced filtering allows specific words, locations and sources to be searched. The lower part of the panel displays concrete reports with the name of the source, title of the report and a brief sample of the text. The entire report is given an exact time and location stamp for the source. The entire panel can be hidden by clicking on the side arrow, thereby expanding the map field.

News is another sub-page and displays a list of aggregated news in the form of a blog. The page has a non-spatial character and serves exclusively as a list of already verified reports stored in the database. This is an extended version of the sidebar located to the right of the map application. The website's structure shown in Figure 7 is again a simple design to permit the best possible user navigation. As with other parts of the site, no banners or widgets are included.

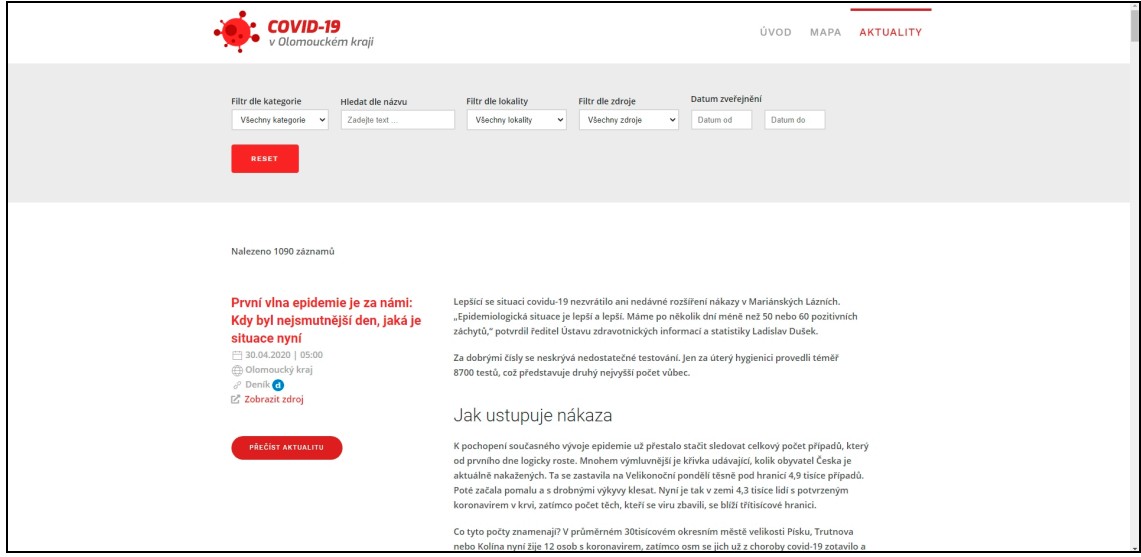

**Figure 7.** News web sub-site.

Extended functionality features advanced data filtering and options for displaying reports. The filter is located at the top of the website, where users can select news according to their preferred category (all, social networks, news reports and state authorities), location (spatial location of news) or

a specific source. The timestamps for each post allow individual reports to be selected according to the date of publication or a user-selectable period. All aggregated records are archived in the database and thus are traceable. A keyword search option using a text search engine is also available.

The main page element contains the reports themselves. Unlike the panel in the map application, a complete transcript of the report is displayed. Users are not linked to the original source portal; textual information is instead directly displayed in the described environment. Photos, images or other visualizations placed in the original report are also displayed in the environment. A timestamp, the source and reference to the original report, and the region for which the report is relevant are indicated.

### 5.2. Cartography

Cartographic accuracy of the methods used to visualize data in the map application was a key criterion in project implementation. Many of the existing map portals listed in Section 3.2 suffer from a series of cartographic transgressions. The most common mistake is incorrect application of the choropleth map. According to Voženílek and Kaňok [25], the main feature of a choropleth map is the representation of a phenomenon expressed by relative values captured in partial territorial units. The correct application of the choropleth method enables individual territorial units to be compared, thereby illustrating the spatial variability of the phenomenon within the processed area. For a correct comparison, it is crucial that the data is relative, and ideally, recalculated to the area of the territorial unit. Recalculation using another characteristic of the territorial unit (e.g., the number of inhabitants) is also acceptable. A very common and fundamental mistake appearing in most published maps is application of this cartographic method of expression to absolute data [26]; the cartodiagram method is used for that purpose, however.

Wrong application of the choropleth map is very common in other similar oriented solutions. Those use different shades of lightness and darkness to represent intensity as being easy to create, but also easy to get wrong [27]. It is an important point that we must not show same numbers from different sized regions with the same color. It is not the same when is somebody infected in center of the Prague or in some small village in the middle of woodland. There is still one case, but with a different region size and amount of people around. This can cause misunderstandings and potential panic (from the point of view of a small village) or underestimation (from the point of view of Prague).

The choropleth map method was used in the application to map coronavirus cases in the Olomouc region. The absolute numbers of persons who test positive for coronavirus in the subregion are processed in proportion to 10,000 inhabitants. The given number of inhabitants is logically selected according to the population in the marked areas of the Olomouc region. The resulting relative values are grouped into five intervals, which are distinguished from each other according to an increasing saturation scale of red (Figure 8).

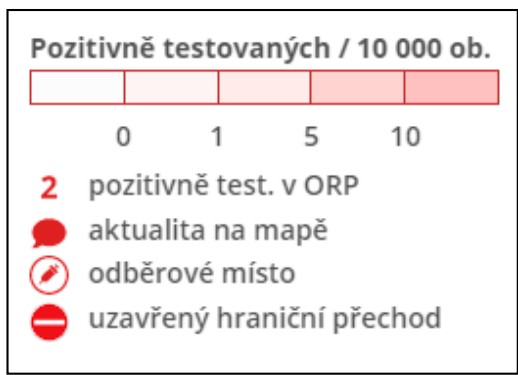

**Figure 8.** Graphical design of the legend.

Using the numerical description located directly on the map, the current absolute value of the number of infected in the given sub-region is expressed. This ensures the greatest possible information value of the map and the possibility to compare the penetration rate between areas.

As the application starts up, spatial information with point layers of closed border crossings and sampling points in the region are displayed by default. The thematically selected symbolic characters are the exact center of the reference point. Their design respects all the general aspects of cartographic features according to Freitag [28]. The information value of the subscription point character is increased with a pop-up window displayed after a user clicks on a specific character. It contains information about the place of collection with an exact address.

After activating the news layer in the map, characters with a numerical description representing specific aggregated news, spatially related news and their exact number are displayed. When users operate in basic mode, this information is associated with a single location in a particular sub-region. As the zoom level increases, the characters separate into their original positions. At the largest possible approximation, aggregation begins in places that have only published more than one report at the given coordinates. This cluster can be divided into separate characters representing specific reports. This feature relates to a specific location according to a rule. When the user clicks, a pop-up window appears with text displaying the title of the report, a timestamp and a link to the original report (Figure 9).

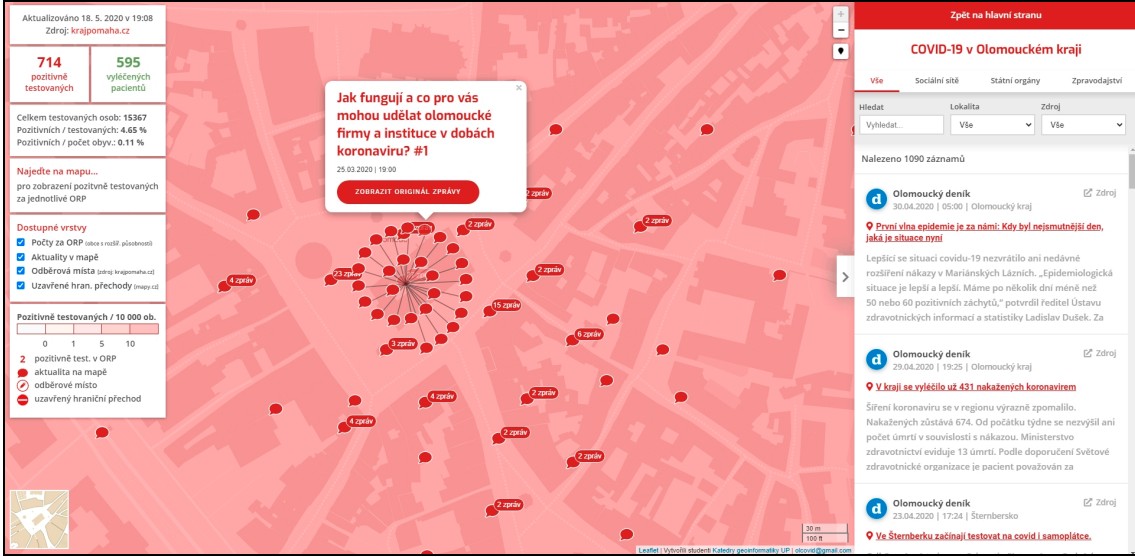

**Figure 9.** Sample of the final version of news clustering in the map.

## 6. Discussion

This project resulted in a completely new portal for visualizing the coronavirus pandemic in the Olomouc Region. The project was clearly prepared and published on the university domain of the Department of Geoinformatics at Palacký University Olomouc (https://gis.upol.cz/covid/). When the platform's interface was developed, simple and instinctive operation was provided, now evidenced by its relatively high website traffic. Since the public launch of the application on 16 March 2020, the web pages have been viewed over 270,000 times, with users spending almost two minutes on average at the site in a single session (see Figure 10). Traffic peaked on the fourth day after launch with a total of 45,000 web views and approximately 15,000 registrations of unique visitors. The numbers then gradually stabilized during the first week to an average of 10,000 access numbers per day, and later, 2000 to 5000 users per day.

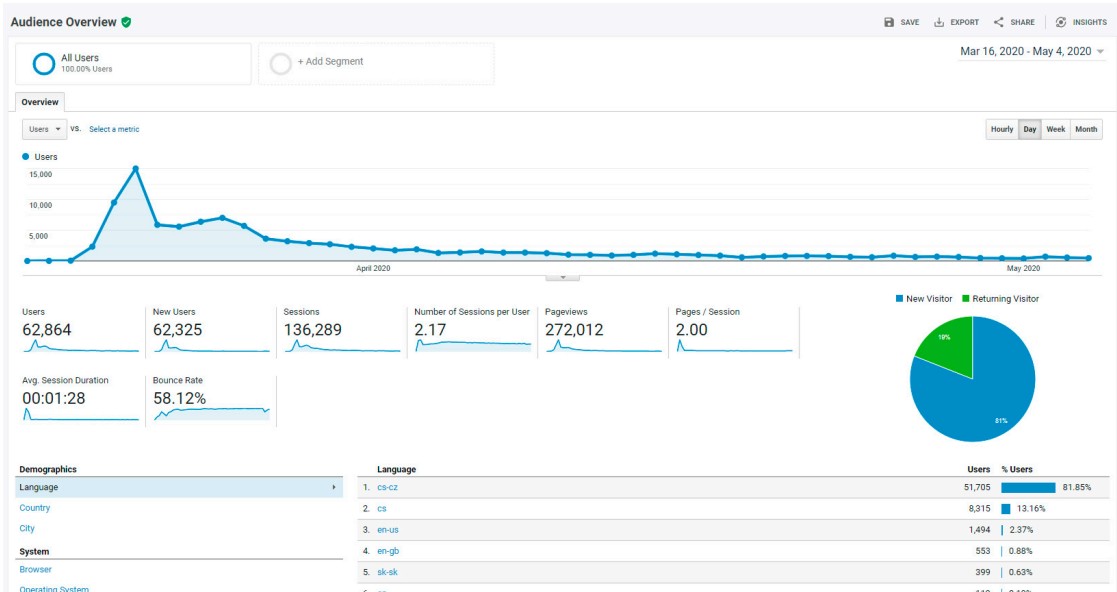

**Figure 10.** The most significant increase of visitors in time according to Google Analytics tables.

More than 50% of all visits to the site were from within the Olomouc region, whose citizens gained the most value from the information. The application addressed the general public, the media, and experts from various fields. The population certainly increased its awareness of the use of GIS technologies and the Palacký University Olomouc, and thus also of the Department of Geoinformatics. The application was also appreciated widely, being seen in over 30 media outlets, including the main national news portals (Figure 11), not only in the regional and national print and online media. The project also received a positive response from regional crisis staff.

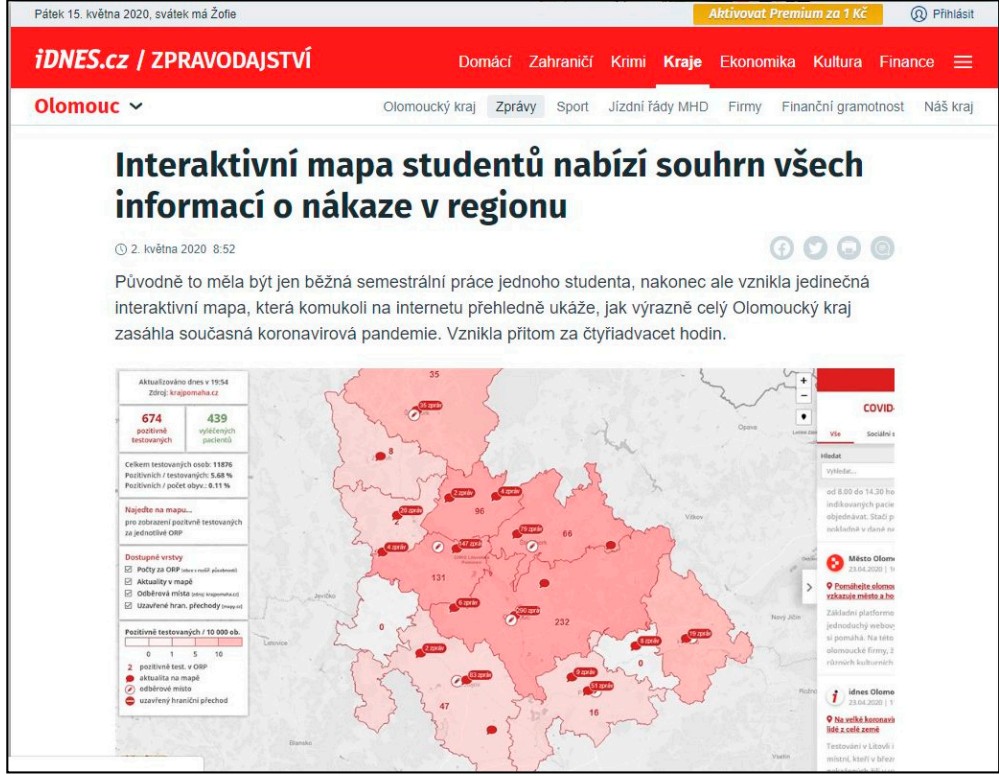

**Figure 11.** Description of the COVID-map project on idnes.cz, a popular Czech media portal.

The resulting application was created explicitly in the Czech language for the reasons of using purely Czech local resources and the regional character of the COVID-map. The IT solution itself is open source and therefore fully transferable and adaptable to custom requirements. However, since the data sources are non-uniform, it would first be necessary to completely modify the settings of the input data aggregator if the application is ported to another area.

The platform has many benefits. It helps raise awareness in citizens within the target area and supports decision-making for actions. It effectively combines various media sources (newspapers, news reports, tweets, etc.) into a single format while also offering spatial visualization of information using modern and interactive technology. It also supports the prestige of the Czech geoinformatics scene and the desire of society to move forward with Industry 4.0. The proposed system emerged as an ad hoc solution to respond to acute needs. However, this approach also has the potential for improvement. Of course, an algorithm is still available, for example, to fully automate the collection of information or advanced spatial localization with an emphasis on address points. Progressive development in user potential could also be generated by increasing the interactivity of the client-side interface (alternative characters, colors, navigation). However, the greatest benefit of the present study lies in its background, where rigorous data collection from numerous sources in a variety of formats is executed. A large part of the work logically deals with unifying and subsequently visualizing the data in the map environment, where they are clearly aggregated and localized.

Practical use of the solution has already been proven in a crisis situation, especially in its unique method of centrally collecting information and the benefits from spatial orientation in decision-making and information retrieval. The contribution of the system has been lauded both by members of administrative units and the public, which reflects direct attendance. After three months of running, the import and validation process was stopped due to situation calm. The map is still available, but data were not updated since the 18th of May, while user traffic was at the minimum level a few days before. Since the application was fully implemented, it is immediately ready for future re-usage at any time. It does not require any further relevant maintenance. From a technical point of view WordPress only requires updates, but this is an automatized process. The biggest variable depends on a personnel issue for manual processes to maintain import channels/sources and verify news. In the future, a suitable implementation may see the map application (or similar) directly connected to the crisis management system to support decision-making, or, for example, to information channels to raise the awareness of the population. The application was designed as universal and easily-expandable. There are minor technical requirements (php7, MySQL5+, SSL) for deployment by other agencies in other regions. A well performing server is crucial. In fact, commercial webhosting with scalable parameters or an oversized server is required to eliminate simultaneous visitors' peaks. Replicability is limited by input sources only. Formal and data issues could be restricted in other regions. The application is strictly dependent on formats of input sources. Not all origin-data formats are supported, as custom channels will require custom import. The biggest limitation could be legal parameters–copyright, General Data Protection Regulation (GDPR) and some local laws may prohibit the usage of certain data without permission (e.g., from social networks, personal blogs, etc.). From a cartographical point of view, a correct coordinate assignment is fundamental. Based on our experiences, users with a cartographical/geographical knowledge are recommended for the verification process. The map application handles with a standardized coordinate format WGS84 (latitude, longitude). Therefore, no visualization issues are expected. Currently, the source code is not available online for free download because of a third-party plugin which does not allow sharing. Authors working on update which will replace it by a custom solution. Following an open source license requirements, it will be available on GitHub.

According to historical experience and modern forecasts, a similar pandemic can be expected once every 100 years. Although the authors have made considerable efforts to develop the COVID-map application, the application has already proven itself and is ready to be used for other, similar purposes. The authors strongly believe and hope the application will no longer be essential in the near future.

**Author Contributions:** Investigation, J.K. (Jakub Konicek); Methodology, J.K. (Jakub Konicek); Project administration, R.N.; Resources, T.B.; Supervision, R.N.; Validation, T.B.; Visualization, J.K. (Jakub Kaplan); Writing—original draft, T.N. and J.K. (Jakub Kaplan); Writing—review & editing, T.N. All authors have read and agreed to the published version of the manuscript.

**Funding:** This research was funded by Internal Grant Agency of Palacký University Olomouc [IGA_PrF_2020_027]. The APC was funded by Internal Grant Agency of Palacký University Olomouc.

**Acknowledgments:** This paper was created under the project "Advanced application of geospatial technologies for spatial analysis, modeling, and visualization of the phenomena of the real world" (IGA_PrF_2020_027) with the support of Internal Grant Agency of Palacký University Olomouc.

**Conflicts of Interest:** The authors declare no conflict of interest.

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
