# Peer review of "Non-Spatial Data towards Spatially Located News about COVID-19: A Semi-Automated Aggregator of Pandemic Data from (Social) Media within the Olomouc Region, Czechia"

_data, 2020_

Round 1

Reviewer 1 Report

Dear colleagues, 

thank you very much for this valuable contribution. It is important to show the corss-domain behaviour of maps and their impact on our decisions, especially in combination with high velocity data. 

You have a clear description of your motivation and try to give an insight on the current conditions of map applications and data aggregation before you come to your specific materials and methods. For my point of view the "current conditions" could be shortened, because the given situation is NOT comprehensive. You will not be able to list all important tools, but you could give examples. 

In "materials and methods" you describe the data sources, technology, administration and agile development. Whereas on one hand the "administration" and "agile development" provides valueable insight, on the other hand those points of view also overload the reader, because most of the technical readers could have left the paper already due to the introduction with focus on COVID-19.
My proposal is ...
- to hint the reader in the introduction part that these aspects (agile development for the webmapping design procedure) will come up. 
- to shorten the "administration" chapter. 
- to enhance the "agile development" chapter by its impact on the webmap creation process. 

In addition, when you make use of agile development, the importance and role of "design thinking" (e.g. https://www.springer.com/gp/book/9783642137563) could be highlighted as preliminary step. 

I like the discussion and highlighting of absolute versus relative numbers and the use case of choropleth maps. The cartography section defines the end product and is therefore one of the most important sections. You should extend this section, especially with a discussion on the impact of the "signs" used. E.g. absolute numbers have their impact on panicking citizens and therefore where an important factor for the dramatic communication for COVID-19.

I am missing a clear distinction between GIS and MAP in the beginning of the paper. I have the impression that GIS and Map are used as synonyms. 

I am missing a statement for the role and importance of topographic base data and basemaps. Is this geometric information needed to be authoritative? If not, why?

I am missing a link to the UN-GGIM and their Global Statistical Geospatial Framework (http://ggim.un.org/meetings/GGIM-committee/9th-Session/documents/The_GSGF.pdf) in the section "Data Agregators". 

I am missing a link and exploration of GISCO (https://ec.europa.eu/eurostat/web/gisco) in the section "Data Agregators", because this service of the European Commission is doing those dataintegrations all the time. 

Author Response

Dear reviewer, thank you for your feedback. We have tried to incorporate all your comments. Generally, we skipped some parts. Our responses are below:

In "materials and methods" you describe the data sources, technology, administration and agile development. Whereas on one hand the "administration" and "agile development" provides valueable insight, on the other hand those points of view also overload the reader, because most of the technical readers could have left the paper already due to the introduction with focus on COVID-19. 
My proposal is ...
- to hint the reader in the introduction part that these aspects (agile development for the webmapping design procedure) will come up. 
- to shorten the "administration" chapter. 
- to enhance the "agile development" chapter by its impact on the webmap creation process. 

Thank you for idea. Agile development is mentioned in introduction, administration chapter was shortened, agile development was revised as well.

In addition, when you make use of agile development, the importance and role of "design thinking" (e.g. https://www.springer.com/gp/book/9783642137563) could be highlighted as preliminary step. 

Thank you for a recommendation, we added ideas from a publication into chapter about developement.

I like the discussion and highlighting of absolute versus relative numbers and the use case of choropleth maps. The cartography section defines the end product and is therefore one of the most important sections. You should extend this section, especially with a discussion on the impact of the "signs" used. E.g. absolute numbers have their impact on panicking citizens and therefore where an important factor for the dramatic communication for COVID-19

We added mentioned thoughts into cartography chapter.

I am missing a clear distinction between GIS and MAP in the beginning of the paper. I have the impression that GIS and Map are used as synonyms. 

Definitelly they are not synonyms. Added into chapter 2.2  : While GIS is a complex software for spatial data input, analysis and visualization, map is an output generated by GIS.

I am missing a statement for the role and importance of topographic base data and basemaps. Is this geometric information needed to be authoritative? If not, why?

Added into chapter 5.1

I am missing a link to the UN-GGIM and their Global Statistical Geospatial Framework (http://ggim.un.org/meetings/GGIM-committee/9th-Session/documents/The_GSGF.pdf) in the section "Data Agregators". 

Added into chapter 3.3

I am missing a link and exploration of GISCO (https://ec.europa.eu/eurostat/web/gisco) in the section "Data Agregators", because this service of the European Commission is doing those dataintegrations all the time. 

Added into chapter 3.3 as well

Reviewer 2 Report

The authors of the paper have developed a mobile responsive web application that aggregates non-spatial and geospatial data on the new COVID-19 from different sources in a region of Czech Republic. This research is a good contribution for scientists who would like to develop a similar tool for tracking such information. The paper is well structured –though poorly referenced-- and fluid. It raises very important questions and issues. Below are a few major and minor comments, that could help to improve it.

Major comments

- Can the authors discuss the validation process for data to be published on their portal, in particular for social network data? In other words, what makes, according to them, a source of information reliable or not? How did the authors choose between two different values of the same indicator coming from two different sources? How were selected the persons in charge of the validation? How will the validation process be maintained in the future, if the pandemic is still active during the coming years?

- Can the authors discuss in more detail the replicability of their application by other scientists or engineers in other geographic zones and at other geographical scales?

- Can the authors indicate how it is feasible to maintain this application in future (technical maintenance and governance)?

- Chapter 4 is too long in my view. It looks sometimes more like a students’ project report than a scientific paper. The authors could reduce this chapter substantively (e.g., lines 250-335), focus less on the various development iterations and keep the essence

- Literature review is weak (22 references), is composed in majority of links to online documents. It would also benefit from having more recent references

- When I tried to open the COVID web application from Chrome and Firefox (https://gis.upol.cz/covid/) the web page was empty

Minor comments

- In the Supplementary Material section, I did not see the contribution of each author to this research

- Where is the source code of the application available? It could be useful for other researchers who would like to replicate this application in their geographical area. For example, is a Github repository available and where?

- Line 98: “Research of the current conditions” could be reformulated, e.g. “Research background”, or “State of the art”

- Line 127: which Ministry of Health is mentioned here?

- Lines 139-143: the Canadian application is not referenced and the application from New-Zealand is referenced by the Canadian reference

- Figure 10: it would be good to know precisely which statistics are available, as the application interface is in Czech language

Author Response

Dear reviewer, thank you for your feedback. We have tried to incorporate all your comments. Generally, we skipped some parts. Our responses are below:

- Can the authors discuss the validation process for data to be published on their portal, in particular for social network data? In other words, what makes, according to them, a source of information reliable or not? How did the authors choose between two different values of the same indicator coming from two different sources? How were selected the persons in charge of the validation? How will the validation process be maintained in the future, if the pandemic is still active during the coming years?

Added to chapter 4.3.3 and 6

- Can the authors discuss in more detail the replicability of their application by other scientists or engineers in other geographic zones and at other geographical scales?

Added to chapter 6: no technical or cartographical issues are expected, but Formal and data issues could restrict in other regions

- Can the authors indicate how it is feasible to maintain this application in future (technical maintenance and governance)?

Added to chapter 6: The pilot study does not require any further relevant maintenance. Since it is fully implemented, it is ready to use again immediately. From technical point of view only WordPress requires update, but it is automatized process. The biggest deal is personal - to maintain import channels/sources and verify news.

- Chapter 4 is too long in my view. It looks sometimes more like a students’ project report than a scientific paper. The authors could reduce this chapter substantively (e.g., lines 250-335), focus less on the various development iterations and keep the essence

The chapter was shortened and instantiated, thank you for comments.

- Literature review is weak (22 references), is composed in majority of links to online documents. It would also benefit from having more recent references

We added few others references

- When I tried to open the COVID web application from Chrome and Firefox (https://gis.upol.cz/covid/) the web page was empty – it works fine

We are sorry, all foreing users were blocked by server setup. Currently, we block robots only, so now the URL is available from abroad.

- In the Supplementary Material section, I did not see the contribution of each author to this research

Investigation, Jakub Konicek; Methodology, Jakub Konicek; Project administration, Rostislav Netek; Resources, Tomas Burian; Supervision, Rostislav Netek; Validation, Tomas Burian; Visualization, Jakub Kaplan; Writing – original draft, Tereza Novakova and Jakub Kaplan; Writing – review & editing, Tereza Novakova.

- Where is the source code of the application available? It could be useful for other researchers who would like to replicate this application in their geographical area. For example, is a Github repository available and where?

That´s a good question. Honestly, we recieved the same question from anothers region which were interested for deployment. Currently nowhere. The reason is that we used third-party plugin (for save some time at the begining) which not allows to share source code. We are curently working on it. We would like to fix it and deploy an updated version. It will be fully open source, it will be on GitHub within one-two months.

- Line 98: “Research of the current conditions” could be reformulated, e.g. “Research background”, or “State of the art”

- Shortened, thank you for an advice

- Line 127: which Ministry of Health is mentioned here?

It is a Swiss Ministry of Health, we added it to the text

- Lines 139-143: the Canadian application is not referenced and the application from New-Zealand is referenced by the Canadian reference

We are sorry for that, a mistake was mended

- Figure 10: it would be good to know precisely which statistics are available, as the application interface is in Czech language

Updated with english interface

Round 2

Reviewer 1 Report

Thank your for embedding my considerations and a big thank for publishing your research.

Reviewer 2 Report

I am fine with the corrections done by the authors. In my view the paper can be published.